# SPARC Is Highly Expressed in Young Skin and Promotes Extracellular Matrix Integrity in Fibroblasts via the TGF-β Signaling Pathway

**DOI:** 10.3390/ijms241512179

**Published:** 2023-07-29

**Authors:** Seung Min Ham, Min Ji Song, Hyun-Sun Yoon, Dong Hun Lee, Jin Ho Chung, Seung-Taek Lee

**Affiliations:** 1Department of Biochemistry, College of Life Science and Biotechnology, Yonsei University, Seoul 03722, Republic of Korea; mongsil1115@naver.com; 2Department of Dermatology, Seoul National University College of Medicine, Seoul 03080, Republic of Korea; minjisong@snu.ac.kr (M.J.S.); hsyoon79@gmail.com (H.-S.Y.); ivymed27@snu.ac.kr (D.H.L.); jhchung@snu.ac.kr (J.H.C.); 3Laboratory of Cutaneous Aging Research, Biomedical Research Institute, Seoul National University Hospital, Seoul 03080, Republic of Korea; 4Institute of Human-Environment Interface Biology, Seoul National University, Seoul 03080, Republic of Korea; 5Department of Dermatology, Seoul National University Boramae Hospital, Seoul 07061, Republic of Korea; 6Institute on Aging, Seoul National University, Seoul 03080, Republic of Korea

**Keywords:** extracellular matrix, fibroblast, MMP-1, skin aging, SPARC, TGF-β, type I collagen

## Abstract

The matricellular secreted protein acidic and rich in cysteine (SPARC; also known as osteonectin), is involved in the regulation of extracellular matrix (ECM) synthesis, cell-ECM interactions, and bone mineralization. We found decreased SPARC expression in aged skin. Incubating foreskin fibroblasts with recombinant human SPARC led to increased type I collagen production and decreased matrix metalloproteinase-1 (MMP-1) secretion at the protein and mRNA levels. In a three-dimensional culture of foreskin fibroblasts mimicking the dermis, SPARC significantly increased the synthesis of type I collagen and decreased its degradation. In addition, SPARC also induced receptor-regulated SMAD (R-SMAD) phosphorylation. An inhibitor of transforming growth factor-beta (TGF-β) receptor type 1 reversed the SPARC-induced increase in type I collagen and decrease in MMP-1, and decreased SPARC-induced R-SMAD phosphorylation. Transcriptome analysis revealed that SPARC modulated expression of genes involved in ECM synthesis and regulation in fibroblasts. RT-qPCR confirmed that a subset of differentially expressed genes is induced by SPARC. These results indicated that SPARC enhanced ECM integrity by activating the TGF-β signaling pathway in fibroblasts. We inferred that the decline in SPARC expression in aged skin contributes to process of skin aging by negatively affecting ECM integrity in fibroblasts.

## 1. Introduction

Secreted protein acidic and rich in cysteine (SPARC; also known as BM-40 and osteonectin) is a matricellular glycoprotein that regulates interactions between cells and the extracellular matrix (ECM). This glycoprotein has acidic, follistatin-like, and extracellular matrix domains that enable direct coupling with ECM proteins [1]. Although SPARC is not a structural component of the ECM, it is involved in ECM remodeling and tissue repair by associating with ECM molecules such as collagen, which is the main ECM protein. Furthermore, SPARC also interacts with secreted growth factors and cell surface receptors involved in angiogenesis, wound healing, cell migration and invasion, cell proliferation, and tissue remodeling. 

SPARC is found in human skin, bone, and lungs, and is expressed by human fibroblasts, osteoblasts, and endothelial cells [2,3,4,5]. SPARC promotes fibrous tissue formation by modulating fibroblast activity. SPARC is abundantly expressed in fibrotic disorders such as systemic sclerosis, pulmonary, and renal interstitial fibrosis [6,7,8,9]. Knockdown of SPARC attenuates fibrosis by reducing collagen expression [10,11,12], and SPARC plays important roles in the pathogenesis of osteoporosis because it regulates bone formation and remodeling by modulating the activities of osteoblasts and osteoclasts [13]. SPARC decreases type IV collagen levels and increases the expression of matrix metalloproteins (MMPs) that break down ECM components, thus modulating the tumor microenvironment and promoting tumor growth, invasion, metastasis, and angiogenesis [14,15,16,17].

Transforming growth factor-β (TGF-β) is a multifunctional cytokine involved in physiological processes such as growth, differentiation, and proliferation in several cell types [18,19]. TGF-β activates receptor-regulated mothers against decapentaplegic (R-SMADs) by binding to TGF-β receptors (TGFBRs), then induces collagen synthesis and suppresses MMPs [20,21]. SPARC can activate TGF-β signaling by interacting with TGF-β receptor type 2 (TGFBR2) that induces SMAD2 phosphorylation [22,23,24]. Moreover, TGF-β can stimulate the expression of SPARC in fibroblasts, keratinocytes, smooth muscle, and endothelial cells [25,26,27,28]. 

SPARC also promotes and reduces ECM integrity [14,29]. We quantified SPARC in young and elderly sun-protected skin and analyzed its effects on type I collagen and MMP-1 at the protein and mRNA levels in two-dimensional (2D) and 3D cultured human fibroblasts to determine the capacity of SPARC to modulate the ECM in human skin. We explored and validated the molecular mechanism through which SPARC regulates type I collagen and MMP-1 expression in fibroblasts. We also assessed differentially expressed genes (DEGs) and significantly changing gene ontologies (GOs) by SPARC in fibroblasts. Our results suggested that SPARC plays a role in intrinsic skin aging via ECM regulation.

## 2. Results 

### 2.1. Expression of SPARC Is Reduced in Elderly Skin Tissues

We analyzed ultraviolet (UV)-protected buttock skin tissues from young and elderly individuals using quantitative RT-PCR and immunohistochemical (IHC) staining to identify age-related changes in SPARC in human skin. Levels of *SPARC* mRNA and IHC-stained protein were both significantly lower in elderly, than in young skin tissues (Figure 1A,B). We also detected SPARC signals particularly in the basal layer of the epidermis, and in fibroblasts and endothelial cells of the dermis. Moreover, a significant decrease in collagen intensity was observed in elderly human skin tissues when compared with young tissues (Appendix A).

### 2.2. SPARC Increased Type I Collagen and Reduced MMP-1 in Human Skin Fibroblasts

We used western blotting to analyze the expression of type I collagen and MMP-1 in foreskin fibroblasts incubated with or without SPARC. We expressed a recombinant SPARC polypeptide by constructing a vector expressing human SPARC with a C-terminal His tag and stably transfecting it into HEK293 cells. His-tagged SPARC was purified from the conditioned medium of cells overexpressing the SPARC polypeptide by affinity chromatography using Ni^2+^-NTA resin (Appendix A). Treatment with SPARC (0–8 μg/mL) resulted in a dose-dependent up-regulation of type I collagen and down-regulation of MMP-1. Notably, saturation was observed at a concentration of 2 μg/mL SPARC or higher (Figure 2A). In addition, SPARC (2 μg/mL) increased type I collagen and decreased MMP-1 secretion in fibroblasts to levels almost similar to those in cells incubated with TGF-β1 (3 ng/mL) (Figure 2B).

We explored whether changes in the secretion of type I collagen and MMP-1 induced by SPARC were regulated at the mRNA level. The results of conventional and quantitative RT-PCR analyses showed that SPARC significantly increased levels of *COL1A1* and *COL1A2* mRNA and decreased those of *MMP-1* mRNA in fibroblasts (Figure 3).

### 2.3. SPARC Increased Secretion of Type I Collagen and Decreased Its Degradation in 3D Cultured Fibroblasts 

We confirmed the effects of SPARC on ECM integrity by analyzing changes in SPARC-induced type I collagen and MMP-1 expression in fibroblasts embedded in a collagen matrix (3D culture) to mimic the dermis in vivo. Immunofluorescence (IF) staining of fibroblasts using antibodies specific to the N-terminal pro-peptide of type I collagen (COL1A1) and three-quarter fragment of type I collagen (Type I collagen cleavage site) revealed that SPARC treatment led to increased synthesis of type I collagen and decreased degradation of type I collagen (Figure 4). The synthesized type I procollagen was observed in the perinuclear region, likely corresponding to the endoplasmic reticulum (ER), while the degraded type I collagen was detected in the pericellular area. These data strongly indicate the critical role of SPARC in maintaining ECM integrity within the dermis of the skin.

### 2.4. SPARC Enhances the Activity of R-SMADs and TGF-β Receptors 

We investigated the activation of R-SMADs, SMAD2 and SMAD3 in foreskin fibroblasts to determine involvement of the TGF-β signaling pathway in SPARC-mediated changes in type I collagen and MMP-1 expression. Both SPARC and TGF-β1 induced SMAD2 and SMAD3 phosphorylation (Figure 5A). We then investigated the effects of SB431523, an inhibitor of TGF-β receptor type 1 (TGFBR1), on SPARC-induced SMAD2 activation and type I collagen as well as MMP-1 secretion in foreskin fibroblasts. SB431542 blocked the SPARC-induced SMAD2 phosphorylation (Figure 5B) and reversed the SPARC-induced increase in type I collagen secretion and decrease in MMP-1 secretion (Figure 5C).

We assessed SPARC-induced time-dependent changes in SMAD2 phosphorylation in foreskin fibroblasts incubated with either SPARC or TGF-β1 or both. The amount of SMAD2 phosphorylation peaked at 30 min after incubation with either SPARC or TGF-β1 or both, but the maximum was higher for the latter (Figure 6). These results indicated that SPARC induced type I collagen expression and suppressed MMP-1 expression by activating TGFBRs and R-SMADs.

### 2.5. SPARC Regulates the Expression of Genes Associated with ECM Integrity in Fibroblasts 

We analyzed transcriptomes by assessing changes in SPARC-regulated gene expression in foreskin fibroblasts. A total of 8017 transcripts exhibited significant differential expression, resulting in the identification of 575 genes as DEGs (Figure 7A). GO terms significantly associated with these DEGs with a higher −log_10_ (adjusted *p*) comprised response to vitamin D (GO: 0033280; *p* = 2.9 × 10^−6^) in the biological process (BP) category; extracellular space (GO: 0005615; *p* = 3.9 × 10^−11^), extracellular region (GO: 0005576; *p* = 5.0 × 10^−7^), extracellular matrix (GO: 0031012; *p* = 8.1 × 10^−6^), and integral component of plasma membrane (GO: 0005887, *p* = 1.1 × 10^−5^) in the cellular component (CC) category, and cytokine activity (GO: 0005125, *p* = 2.9 × 10^−5^) in the molecular function (MF) category (Figure 7B). We selected nine DEGs in these six GO terms, based on the criterion of absolute (log_2_ (fold change)) ≥ 1.5. Eight DEGs upregulated by SPARC were elastin (*ELN*), tetraspanin 2 (*TSPAN2*), cartilage oligomeric matrix protein (*COMP*), olfactomedin 2 (*OLFM2*), Wnt family member 2 (*WNT2*), cellular communication network factor 2 (*CCN2*), tetraspanin-13 (*TSPAN13*), and serine proteinase inhibitor family E member 1 (*SERPINE1*), mainly belonged to the GO terms, extracellular region, extracellular space, and extracellular matrix. The other DEG downregulated by SPARC was the prostaglandin-endoperoxide synthase (*PTGS2*) gene that is often upregulated in inflammation. The expression of these genes in SPARC-treated foreskin fibroblasts (Figure 7C) using quantitative RT-PCR analysis (Appendix A) were consistent with the transcriptomic data (Appendix A). These findings suggested that SPARC significantly affected the expression of many genes involved in enhancing ECM integrity in fibroblasts.

## 3. Discussion

The secreted matricellular glycoprotein SPARC is abundantly expressed during dermal wound repair and in association with angiogenesis [30,31]. The expression of SPARC is decreased in the skin of aged mice, and collagen strength and abundance are reduced in the skin of SPARC-null mice [32]. However, age-related changes in SPARC abundance in human skin have remained unknown. We found significantly lower *SPARC* mRNA abundance and substantially lower immunoreactivity to SPARC in skin tissues from elderly than young participants. 

Changes in the ECM, such as decreased collagen biosynthesis and increased collagen degradation through increased MMP activity are significant during the aging process [33,34,35]. SPARC is involved in ECM assembly and composition, including pro-collagen processing and collagen fibril formation [29]. Overexpressed SPARC promotes collagen production and ECM synthesis [36], whereas SPARC knockdown has the opposite effects [11] in fibroblasts. SPARC modulates cell proliferation, cell adhesion, and invasion through interaction with ECM components, cell surface receptors, and growth factors [37,38]. We speculate that SPARC plays a key role in intrinsic skin aging by modulating the ECM based on these previous and the present findings.

The role of SPARC in enhancing collagen levels in several cell types has been established. For example, SPARC increases the secretion of type I collagen in cardiac fibroblasts and upregulates mRNA and protein levels of type IV and VII collagen in human dermal fibroblasts and epidermal keratinocytes [24,39]. Overexpressed SPARC increases the production of type I and III collagen in keloid fibroblasts [36], whereas its knockdown reduces type I collagen mRNA and protein levels in dermal fibroblasts and other cell types [10,12,40,41,42], and decreases type II and III collagen expression in keloids and dermal fibroblasts from patients with systemic sclerosis [11,36]. However, whether SPARC induces changes in MMP-1 expression has remained unknown. Here, we revealed that SPARC increased the secretion of type I collagen and decreased the secretion of MMP-1 in foreskin fibroblasts. We also found that SPARC upregulated *COL1A1* and *COL1A2* mRNA, and downregulated *MMP-1* mRNA, indicating that it modulates type I collagen and MMP-1 at the transcriptional level.

The skin consists of epidermal, dermal, and hypodermal layers that form a 3D space. Comparative analysis of total collagen between wild-type and SPARC-null cultures in a 3D environment formed by fibrin gel did not show significant differences in collagen production, except for old fibroblasts [43]. However, SPARC-induced collagen production and degradation in human fibroblasts under 3D conditions that mimic those in vivo have not been investigated. We determined that SPARC increased the biosynthesis of type I collagen and reduced its cleavage by MMP-1 in 3D-cultured fibroblasts in a collagen matrix. These results showed that SPARC upregulates type I collagen and downregulates MMP-1 under 2D and 3D conditions. Therefore, we suggest that SPARC is an important factor for improving ECM integrity and slowing intrinsic skin aging.

The TGF-β signaling pathway is activated by SPARC. For instance, SPARC induces phosphorylation and the nuclear localization of R-SMADs in cardiac fibroblasts and Mv1Lu cells [23,24]. Knockdown of SPARC attenuates the fibrotic effect induced by TGF-β1 at least in part, by inactivating the SMAD2/3 pathway in human pterygium fibroblasts [10]. Here, we also found that SPARC activates SMAD2 and SMAD3 in foreskin fibroblasts. 

Knockdown of TGFBR2 impairs TGF-β signaling induced by SPARC in esophageal adenocarcinoma cells [44]. Notably, SPARC does not directly bind to either TGF-β1 or TGFBR2 that is TGF-β1-free but interacts with TGF-β1 and TGFBR complex only when TGF-β1 induces a conformational change in TGFBR2 [22]. Incubating fibroblasts with SB43152, a potent and selective inhibitor of TGFBR1, suppressed SPARC-induced SMAD2 phosphorylation and type I collagen production, but increased MMP-1 secretion.

Our time-course analysis revealed that the activation of SMAD2 mediated by either SPARC or TGF-β1 peaked at 30 min. Furthermore, co-incubation with SPARC and TGF-β1 activated more SMAD2, although it also peaked at 30 min. We considered two possibilities for the mechanism of SPARC-mediated TGFBR activation based on our findings. One is that SPARC directly binds to and activates TGFBRs independently of TGF-β1, and the other is that SPARC promotes TGFBR2 interaction with TGF-β1 even at TGF-β1 concentrations that are insufficient to activate TGFBR. However, considering that SPARC does not interact with TGFBRs without TGF-β1 [22], the second hypothesis is the more likely mechanism through which SPARC activates TGF-β/SMAD signaling.

The genes and biological phenomena controlled by SPARC in fibroblasts have not been extensively studied. Therefore, we sequenced RNA from foreskin fibroblasts incubated with or without SPARC. Among many DEGs, the most substantially altered were those that were mainly associated with the six GO terms, response to vitamin D, extracellular space, extracellular region, extracellular matrix, integral component of the plasma membrane, and cytokine activity.

We selected nine DEGs with abundant differential expression in the six GO terms described above. Among them, SPARC upregulated ELN and COMP that are components of the ECM, CCN2, OLFM2, and WNT-2 that are involved in ECM production [45,46,47], the ECM protease inhibitor SERPINE1 [48], as well as TSPAN2 and TSPAN13 that are involved in various processes including tissue differentiation [49,50]. SPARC downregulated PTGS2, which is involved in skin inflammation and photoaging [51]. Moreover, RT-qPCR confirmed that SPARC modulated the selected genes as found by the transcriptomic analysis. Based on our findings, we suggest that SPARC maintains ECM integrity by regulating the expression of several genes in fibroblasts, including collagen and MMP-1, thus providing evidence for a role of SPARC in delaying skin aging.

## 4. Materials and Methods

### 4.1. Antibodies

Anti-pro-collagen α1(1) N-propeptide (pN-Col1 α1) and anti-MMP1 antibodies were obtained as described [52]. We sourced antibodies from their respective suppliers: anti-SPARC (15274-1-AP; Proteintech, Chicago, IL, USA); anti-GAPDH (ab83108; AbClone, (Seoul, Republic of Korea); anti-SMAD2 (#5339), anti-phospho-SMAD2 (#3108), anti-SMAD3 (#9523), and anti-phospho-SMAD3 (#9520) (all from Cell Signaling Technology, Danvers, MA, USA); anti-collagen type I cleavage site (immunoGlobe Antikörpertechnik GmbH, Himmelstadt, Germany); horseradish peroxidase (HRP)-conjugated goat anti-mouse IgG, goat anti-rabbit IgG, Alexa Fluor^®^ 488 goat anti-rabbit IgG (H+L), and rhodamine Red X-conjugated goat anti-mouse IgG (Thermo Fisher Scientific Inc., Waltham, MA, USA) [53].

### 4.2. Collection of Human Skin Tissues

Punch biopsy specimens (8 mm) obtained from the buttocks of young (ages 24, 21, and 28 y) and elderly (ages 87, 79, and 76 y) women without skin disease were fixed in 10% formalin, frozen in liquid nitrogen, or stored at −80 °C. The Institutional Review Board of Seoul National University Hospital approved all procedures involving human participants (Approval no: 1410-134-621). All the women described above provided written informed consent to participate in this study, which complied with the ethical principles enshrined in the Declaration of Helsinki (2013 amendment).

### 4.3. IHC Analysis of SPARC

IHC analysis of formalin-fixed human skin tissues was conducted as described previously [54] with goat anti-SPARC antibody (diluted 1:200) in a humidified chamber at 4 °C for 16 h. Controls were stained with a normal goat IgG antibody without immunoreactivity (data not shown).

### 4.4. Cell Culture

Normal human primary foreskin fibroblasts (Welgene Inc., Gyeongsan, Republic of Korea) were cultured in Dulbecco modified Eagle medium (DMEM; Hyclone, South Logan, UT, USA) supplemented with 10% Gibco fetal bovine serum (FBS; Thermo Fisher Scientific Inc.). HEK293 cells were cultured in DMEM supplemented with 10% Gibco bovine serum (BS; Thermo Fisher Scientific Inc.), 100 U/mL penicillin, and 100 μg/mL streptomycin. All cells were cultured at 37 °C under a 5% CO_2_/95% air atmosphere.

### 4.5. Construction of Human SPARC Expression Vector

We generated pcDNA3.1-SPARC-His encoding full-length human SPARC (GenBank NM_003118.4) with a His-tag on the C-terminal tail as follows. Human full-length SPARC cDNA with a His-tag was amplified by PCR using pOTB7-SPARC (Clone ID_hMU007196; Korea Human Gene Bank, Daejeon, Republic of Korea) as a template and PrimeSTAR GXL polymerase (Takara Bio Inc., Kusatsu, Japan). The forward primer 5′-*GACGCTAGC*ACC**ATG**AGGGCCTGGATCTTC-3′ included an NheI site (italics) and nucleotides 53–82 of GenBank NM_003118.4, containing a start codon (bold). The reverse primer 5′-*CGGGATCC***TTA**ATGATGATGATGATGATGGATCACAAGATCCTTGTCGA TATCCTTCTGC-3′ included a BamHI site (italics), His-tag (underline), a stop codon (bold), and nucleotides 976–943 of GenBank NM_003118.4. The PCR products cleaved with NheI and BamHI were ligated into the NheI and BamHI sites of the pcDNA3.1 plasmid. The accuracy of the construct was confirmed by sequencing. 

### 4.6. Purification of Recombinant Human SPARC-His

Sub-confluent HEK293 cells were transfected with pcDNA3.1-SPARC-His using the calcium phosphate method [55]. Transfected cells were screened with G418 (1200 μg/mL) for two weeks, then HEK293-SPARC-His cells stably expressing SPARC-His were enriched. The cells were expanded in DMEM supplemented with 10% BS, then incubated in serum-free DMEM for 5 days. Proteins from the conditioned medium were precipitated with 80% ammonium sulfate, pelleted, and resuspended in PBS containing 10 mM imidazole. Recombinant human SPARC-His was purified from resuspended pellets using Ni^2+^-NTA agarose (QIAGEN, Hilden, Germany) and dialyzed with PBS as described [56].

### 4.7. Western Blotting

Foreskin fibroblasts were serum deprived for 12 h. The medium was replaced with fresh serum-free medium, and the cells were incubated in the presence of SPARC (2 µg/mL unless defined) or TGF-β1 (3 ng/mL) for 12 h in mRNA analysis or for 24 h in protein analysis. Conditioned media obtained from foreskin fibroblasts were centrifuged at 2000× *g* for 5 min to remove cell debris, washed twice with PBS then harvested in sodium dodecyl sulfate (SDS) sample buffer to analyze type I collagen, MMP-1, and GAPDH. Cells were lysed in radioimmunoprecipitation assay (RIPA) buffer (50 mM Tris-HCl, pH 7.4, 150 mM NaCl, 1% NP-40, 0.5% sodium deoxycholate, 0.1% SDS) with a protease inhibitor cocktail (Calbiochem, San Diego, CA, USA), as described previously [56,57]. All samples were boiled in SDS sample buffer containing 100 mM β-mercaptoethanol for 3 min, then resolved by SDS-polyacrylamide electrophoresis (PAGE). Immunoreactive signals were detected by western blotting.

### 4.8. RNA Extraction and RT-PCR 

Total RNA isolation, cDNA synthesis, and RT-PCR proceeded as described [58,59]. The PCR cycling steps were: 94 °C for 30 s (denaturation), optimized annealing temperatures (Appendix A) for 60 s, and 72 °C for 30 s (extension). The PCR products were resolved by 5% PAGE and visualized by staining with ethidium bromide. Quantitative RT-PCR proceeded at optimal annealing temperatures using QuantiTect SYBR Green PCR kits (QIAGEN) and a QuantStudio 3 Real-Time PCR system (Applied Biosystems, Foster City, CA, USA). Target gene expression was normalized to that of GAPDH in equivalent samples.

### 4.9. Synthesis and Degradation of Type I Collagen in 3D-Culture 

Cells were cultured in a 3D system and IF staining proceeded as described [60] with some modifications. Fibroblasts (1 × 10^6^ cells/mL) were obtained by trypsinization and resuspended in the mixture containing rat tail collagen 1 solution (2.8 mg/mL; Corning Inc., Corning, NY, USA), 5× DMEM, and 10× reconstitution buffer (260 mM NaHCO_3_, 200 mM HEPES, and 50 mM NaOH) at a ratio of 7:2:1. Subsequently, 0.15 mL of the mixture was loaded onto a glass-bottom (35 mm × 10 mm, hole 13 φ) dish (SPL Life Sciences, Pocheon, Republic of Korea) and incubated for 30 min at 37 °C to polymerize gel. 1.5 mL of phenol red-free DMEM with or without SPARC was supplemented, and then collagen-embedding cells were incubated for 24 h at 37 °C under 5% CO_2_ and 95% air. Cells in the 3D culture were fixed by 3.7% paraformaldehyde (PFA) for 30 min and permeabilized using 0.2% Triton-X 100 for 10 min. For nucleus staining, the cells were incubated in Hoechst 33258 (2 μg/mL) for 30 min, blocked with 3% bovine serum albumin (BSA) for 30 min, and then immunostained overnight at 4 °C with rabbit anti-type I collagen cleavage-site antibody (2.5 μg/mL) and mouse anti-pN COL1A1 antibody (1:20). After incubation for 2 h with Alexa Fluor^®^ 488 goat anti-rabbit IgG (H+L) and anti-mouse IgG (H+L) Rhodamine Red-X (1 U/mL), all the images were acquired from a confocal microscope (LSM880; Carl Zeiss, Oberkochen, Germany) with the Zen software (Carl Zeiss) under the same settings. The intensity of fluorescence was measured using ImageJ software (National Institutes of Health, Bethesda, MD, USA).

### 4.10. Transcriptomic Analysis

Foreskin fibroblasts were subjected to transcriptome analysis through RNA sequencing using an Illumina NovaSeq platform (Illumina Inc., San Diego, CA, USA). DEGs were analyzed using R package (TCC v.1.26.0) and identified based on the *q*-value threshold less than 0.05 for correcting errors caused by multiple-testing. *Q*-value was calculated based on the *p*-value using the p.adjust function of R package with default parameter settings. For functional characterization of the DEGs, GO based trend test was carried out using the R package (GOseq) through the Wallenius non-central hypergeometric distribution. Selected genes of *p*-values < 0.05 following the test were regarded it as statistically significant.

### 4.11. Statistical Analysis

All data are shown as the means ± SD of at least three independent experiments. Statistical significance was analyzed using unpaired, two-tailed Student *t*-tests, and *p* < 0.05 was regarded as statistically significant.

## 5. Conclusions

The present study found that SPARC mRNA and protein levels decreased with age in human skin tissues. Recombinant SPARC increased the expression of type I collagen and decreased the expression of MMP-1 in fibroblasts at the mRNA and protein levels. Under 3D culture conditions that mimic the dermis, SPARC promoted the biosynthesis of type I collagen and impeded its degradation. We also found that SPARC activated the TGF-β signaling pathway through R-SMAD phosphorylation in the same way as TGF-β1. Our transcriptomic data indicated that SPARC regulates the expression of genes associated with enhanced ECM integrity in fibroblasts. Based on these results, we suggest that SPARC plays a role in maintaining the integrity of the ECM by regulating ECM components, thus contributing to retarding the process of intrinsic skin aging.

## Figures and Tables

**Figure 1 ijms-24-12179-f001:**
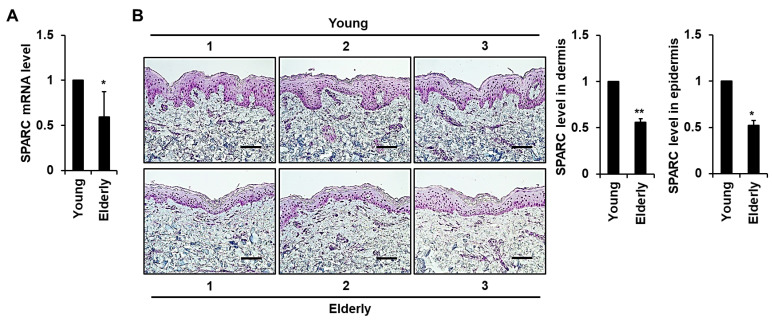
Analysis of SPARC expression in young (ages 24, 21, and 28 y) and elderly (ages 87, 79, and 76 y) human skin tissues. (**A**) Messenger RNA level of *SPARC* in human skin tissues determined by quantitative RT-PCR. Graph shows comparison of *SPARC* mRNA levels between elderly and young human skin tissues. Values are shown as means ± standard deviation (SD) of eight independent experiments. * *p* < 0.05 vs. young tissues. (**B**) IHC detection of SPARC polypeptide in skin tissues from young and elderly participants. Specimens were incubated with SPARC antibody, followed by horseradish peroxidase-conjugated secondary antibody with 3-amino-9-ethylcarbazole, and counterstained with hematoxylin. Graphs show the relative levels of SPARC protein in the dermis and epidermis of elderly and young human skin tissues. Values are shown as means ± SD of three independent experiments. * *p* < 0.05 and ** *p* < 0.01 vs. young tissues. Magnification, ×200. Bar, 100 µm.

**Figure 2 ijms-24-12179-f002:**
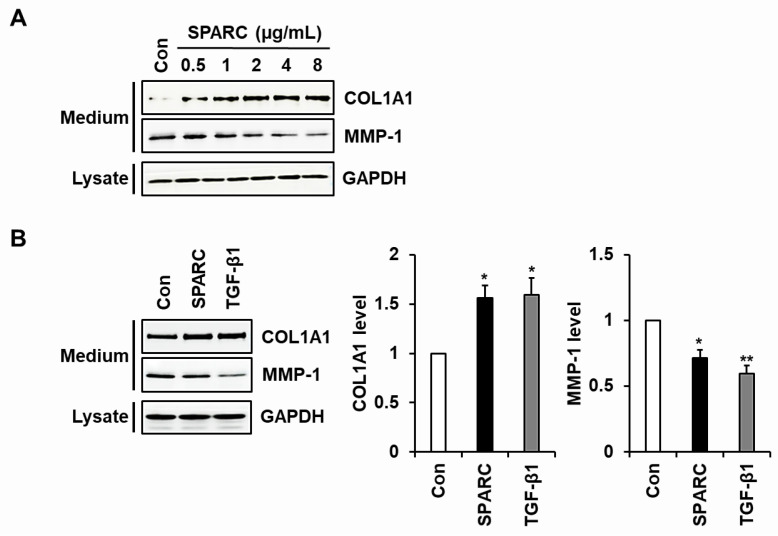
Effects of SPARC on secretion of type I collagen and MMP-1 in fibroblasts. Subconfluent human foreskin fibroblasts were incubated in serum-free DMEM with SPARC (0–8 μg/mL) for (**A**), (2 μg/mL) for (**B**), or TGF-β1 (3 ng/mL) or without (Con) for 24 h. Expression of type I collagen and MMP-1 in conditioned media and of GAPDH in fibroblast lysates was evaluated by western blotting using antibodies against pN-COL1A1, MMP-1, and GAPDH. Graphs show relative levels of COL1A1 and MMP-1 proteins. Values represent means ± SD of three independent experiments. * *p* < 0.05 and ** *p* < 0.01 vs. Con.

**Figure 3 ijms-24-12179-f003:**
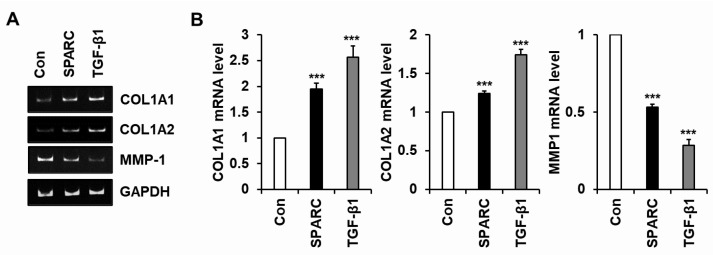
Effects of SPARC on mRNA levels of *COL1A1*, *COL1A2*, and *MMP-1* in fibroblasts. Serum-starved foreskin fibroblasts were incubated for 12 h with SPARC (2 µg/mL) or TGF-β1 (3 ng/mL) or without (Con). Levels of *COL1A1, COL1A2*, and *MMP-1* mRNAs were evaluated using conventional (**A**) and quantitative (**B**) RT-PCR analyses. Values are shown as means ± SD of five independent experiments. *** *p* < 0.001 vs. Con.

**Figure 4 ijms-24-12179-f004:**
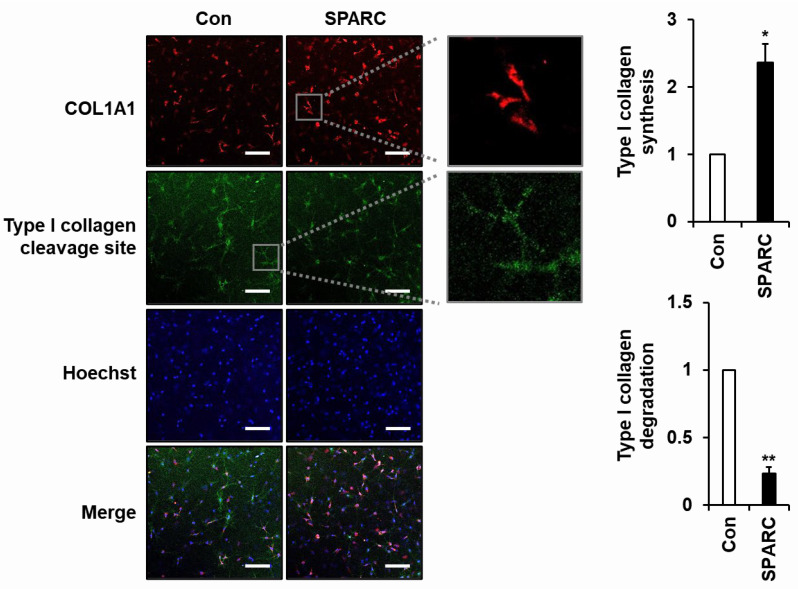
Effects of SPARC on biosynthesis and degradation of type I collagen in 3D cultured fibroblasts. Foreskin fibroblasts were embedded in a type I collagen matrix to mimic the dermis in vivo, then incubated in serum-free and phenol-red-free DMEM without (Con) or with SPARC (2 µg/mL) for 24 h. Foreskin fibroblasts were stained with mouse anti-pN-COL1A1 and Rhodamine Red-conjugated anti-mouse IgG antibodies to detect nascent type I collagen, and with rabbit anti-type I collagen cleavage-site and Alexa Fluor 488-conjugated anti-rabbit IgG antibodies to detect cleavage site of type I collagen. Nuclei were stained with Hoechst 33258. Type I collagen synthesis and degradation were analyzed by confocal fluorescence microscopy. Boxed areas are enlarged to show subcellular images of type I collagen synthesis and type I collagen degradation. Staining intensity of synthesized and degraded type I collagen was quantified using ImageJ software and normalized to nuclear staining intensity. Values are shown as means ± SD of three independent experiments. * *p* < 0.05 and ** *p* < 0.01 vs. Con. Magnification, ×200; bar, 200 µm.

**Figure 5 ijms-24-12179-f005:**
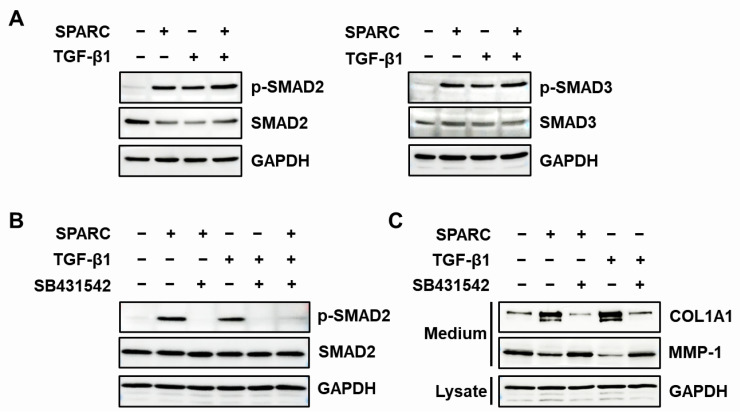
Analysis of TGF-β receptor-regulated R-SMADs activation by SPARC in fibroblasts. (**A**) Foreskin fibroblasts were starved for 12 h, then incubated with SPARC (2 µg/mL) or TGF-β1 (3 ng/mL) for 30 min. Cell lysates were analyzed by western blotting using antibodies against phospho-SMAD2 (p-SMAD2), SMAD2, phospho-SMAD3 (p-SMAD3), SMAD3, and GAPDH. (**B**) Serum-starved foreskin fibroblasts were incubated with SB431542 (10 µM) for 10 min then stimulated with SPARC (2 µg/mL) or TGF-β1 (3 ng/mL) for 30 min. Levels of p-SMAD2, SMAD2, and GAPDH were examined by western blotting using p-SMAD2, SMAD2, and GAPDH antibodies. (**C**) Serum-starved foreskin fibroblasts were incubated with SB431542 (10 µM) and stimulated with SPARC (2 µg/mL) or TGF-β1 (3 ng/mL) for 24 h. Expression of type I collagen and MMP-1 in conditioned media and of GAPDH in cell lysates was evaluated by western blotting using antibodies against pN-COL1A1, MMP-1, and GAPDH.

**Figure 6 ijms-24-12179-f006:**
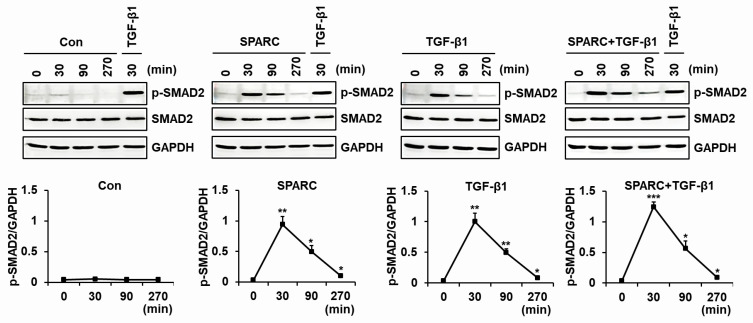
Time course effects of SPARC on SMAD2 phosphorylation in fibroblasts. Serum-starved foreskin fibroblasts were incubated with vehicle (Con), SPARC (2 μg/mL), or TGF-β1 (3 ng/mL) for 0, 30, 90, and 270 min. Cell lysates were analyzed by western blotting using antibodies against p-SMAD2, SMAD2 and GAPDH. Graph shows relative intensity of p-SMAD2/GAPDH in fibroblasts incubated with SPARC or TGF-β1, normalized to that of fibroblasts incubated with TGF-β1 for 30 min. Values are shown as means ± SD of three independent experiments. * *p* < 0.05, ** *p* < 0.01, and *** *p* < 0.001 vs. 0 min.

**Figure 7 ijms-24-12179-f007:**
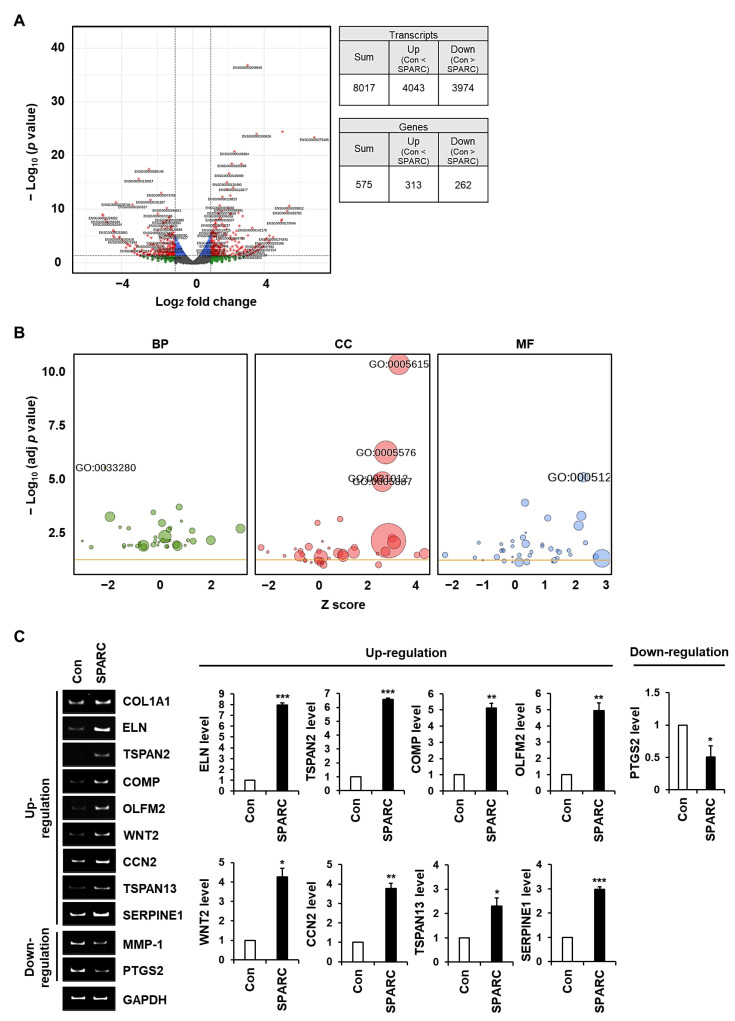
Identification of genes differentially expressed by SPARC in fibroblasts via RNA sequencing. We sequenced mRNA derived from foreskin fibroblasts incubated with SPARC (2 μg/mL) or without (Con) for 12 h. (**A**) DEGs were identified by analyzing three independent sets of RNA mixtures obtained from foreskin fibroblasts incubated without (Con) or with SPARC (2 μg/mL) for 12 h. DEGs were selected based on criteria of absolute (log_2_ (fold change)) ≥ 1.0 and *p*-value < 0.05. (**B**) DEGs were classified into three GO categories: BP, CC, and MF. Significance of GO terms in each category is indicated as -log_10_ (adjusted *p*) values. (**C**) Nine genes selected based on absolute (log_2_ (fold change)) ≥ 1.5 and *p*-value < 0.05. Messenger RNA levels of nine genes, *COL1A1*, *MMP-1*, and *GAPDH*, were evaluated using conventional (**left**) and quantitative (**right**) RT-PCR analyses. Values are shown as means ± SD of three independent experiments. * *p* < 0.05, ** *p* < 0.01, and *** *p* < 0.001 vs. Con.

## Data Availability

Data are contained within the article.

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
