# Peer review of "SPARC Is Highly Expressed in Young Skin and Promotes Extracellular Matrix Integrity in Fibroblasts via the TGF-β Signaling Pathway"

_ijms, 2023, doi:10.3390/ijms241512179_

Round 1

Reviewer 1 Report

Comment on the manuscript ijms-2466791 entitled: “SPARC is highly expressed in young skin and promotes extracellular matrix integrity in fibroblasts via the TGF-signaling pathway: by Seung Min Ham and collaborators.

This manuscript focused on the role of SPARC (Secreted Protein Acidic and Rich in Cysteine, also known as Osteonectin) in skin aging, specifically in relation to the extracellular matrix (ECM) and TGF-β (Transforming Growth Factor-beta) signaling pathway.

Specific comments

1-The authors found that the expression of SPARC decreases in aged skin (Figure 1)

a-In Figure1B the age of the donors should be indicated.

b-the quality of the images in Figure 1B are poor as we hardly can see the staining of SPARC. Consequently, it is difficult to determine whether the signal is indeed decreased in elderly skin. The only difference I can see is that the epidermis in elderly skin is flattened. 

c-Additional staining’s with various markers (vimentin, collagen etc.) are needed to show the age-related changes occurring with age in relation to SPARC staining. 

Immunofluorescence detection will be preferable to provide higher resolution and quantification of the signals.

2- The authors examined the effect of SPARC on the expression of type I collagen and MMP-1 in foreskin fibroblasts. They found that SPARC increased the level of type I collagen and decreased the level of MMP-1 in the fibroblasts, which is consistent with the effects of TGF-β1.

a-Did the authors examine the dose-dependency of the SPARC effect and how did they select the concentration (2 μg/mL) tested?

b-The authors determined the levels of expression of type I collagen and MMP-1 in conditioned media (Figure 2). Further details should be provided in the method section on how the condition media was prepared for western blot analysis.   

3- The authors used a 3D culture of fibroblast to analyze changes in type I collagen and MMP-1 during treatment with SPARC (figure 4).

a-The method section should contain further details on the 3D fibroblast culture.

b-Higher magnification of the immunostaining should be provided in Figure 4 as COL1A1 staining appears localized at the nuclear region. Hence, the overall cellular morphology is difficult to visualize even in images with type 1 collagen cleavage site staining.

4- The authors examined whether the TGF-β signaling pathway is involved in SPARC-mediated changes in type I collagen and MMP-1 expression (Figure 5).

Both SPARC and TGF-β1 induced phosphorylation of R-SMADs (SMAD2 and SMAD3). This is consistent with the canonical TGF-β signaling pathway. Using SB431542, an inhibitor of the TGF-β receptor type 1 (TGFBR1). They show that SPARC is working upstream or at the level of the TGF-β receptor. 

a-The authors suggest that SPARC might be a potential downstream target of TGFβ. They should check by western blot analysis whether fibroblasts treated with TGFβ indeed induce expression of SPARC. This is important, in light that TGFb signaling is activated during the process of aging in the skin.

5-The authors performed transcriptomic analysis to assess changes in SPARC-regulated gene expression in foreskin fibroblasts 

a-The methodology for this analysis is missing

b-The data of this analysis are missing. The authors show only graphs of the GO categories 

Overall, this study is interesting, however, there is an important lack of information the method and result sections. 

Author Response

Thanks for your comment. We have detailed our responses to your comments in this file.

Reviewer 2 Report

The authors investigated the role of SPARC in skin aging. By using primary cell cultures and 3D-culture, they showed that 1) SPARC increased type I collagen and reduced MMP-1 in human skin fibroblasts, 2) SPARC enhanced the activity of R-SMADs and TGF-β receptors, 3) both SPARC and TGF-β1 induced SMAD2 and SMAD3 phosphorylation, 4 ) SPARC regulated the expression of genes associated with ECM integrity in fibroblasts. Taken together, the authors presented a mechanistic cascade how SPARC participates in skin aging. The topic is interesting and most of authors' data support their interpretations.  I have several comments for the authors.

1. The authors stated that Figure 1B showed IHC-stained protein were both significantly lower in elderly, than in young skin tissues. However, the results cannot be clearly observed in Figure 1B. Could the authors use qualitative or semi-qualitative method to validate the statement?   

2. Based on Figure 5B. It seems that SB43a523 enhanced phosphorylation of SPARC-induced SMAD2, rather than blocked phosphorylation of SPARC-induced SMAD2 as the authors stated. Please clarify it.

3. The article may be more interesting for reader if there is a figure summarizing the mechanistic cascades the authors identified. 

Author Response

(The authors gave the same response as above.)

Round 2

Reviewer 1 Report

I am satisfied with the revised manuscript.

Reviewer 2 Report

The authors have revised the manuscript and corrected data based on my comments. I suggest acceptance of the manuscript.